# Characterization of 3D NSCLC Cell Cultures with Fibroblasts or Macrophages for Tumor Microenvironment Studies and Chemotherapy Screening

**DOI:** 10.3390/cells12242790

**Published:** 2023-12-08

**Authors:** Anali del Milagro Bernabe Garnique, Glaucia Maria Machado-Santelli

**Affiliations:** Department of Cell and Developmental Biology, Institute of Biomedical Sciences, University of São Paulo, Ave., Prof, Lineu Prestes, 1524, Cidade Universitária, São Paulo 05508-000, SP, Brazil; anabega@usp.br

**Keywords:** 3D cell culture, spheroids, co-culture, tumor microenvironment, fibroblasts, macrophages

## Abstract

The study of 3D cell culture has increased in recent years as a model that mimics the tumor microenvironment (TME), which is characterized by exhibiting cellular heterogeneity, allowing the modulation of different signaling pathways that enrich this microenvironment. The TME exhibits two main cell populations: cancer-associated fibroblasts (CAFs) and tumor-associated macrophages (TAM). The aim of this study was to investigate 3D cell cultures of non-small cell lung cancer (NSCLC) alone and in combination with short-term cultured dermal fibroblasts (FDH) and with differentiated macrophages of the THP-1 cell line. Homotypic and heterotypic spheroids were morphologically characterized using light microscopy, immunofluorescence and transmission electron microscopy. Cell viability, cycle profiling and migration assay were performed, followed by the evaluation of the effects of some chemotherapeutic and potential compounds on homotypic and heterotypic spheroids. Both homotypic and heterotypic spheroids of NSCLC were generated with fibroblasts or macrophages. Heterotypic spheroids with fibroblast formed faster, while homotypic ones reached larger sizes. Different cell populations were identified based on spheroid zoning, and drug effects varied between spheroid types. Interestingly, heterotypic spheroids with fibroblasts showed similar responses to the treatment with different compounds, despite being smaller. Cellular viability analysis required multiple methods, since the responses varied depending on the spheroid type. Because of this, the complexity of the spheroid should be considered when analyzing compound effects. Overall, this study contributes to our understanding of the behavior and response of NSCLC cells in 3D microenvironments, providing valuable insights for future research and therapeutic development.

## 1. Introduction

Cancer research has increasingly relied on 3D cell culture, specifically spheroids, as a valuable tool for testing and developing new anti-cancer drugs. Unlike traditional 2D cell culture, spheroids more accurately mimic the morphological and physiological characteristics of the in vivo microenvironment, which are critical factors influencing therapeutic efficacy [1]. This conformation allows cells to grow and enables surface receptors to interact with neighboring cells, influencing signal translation and gene expression [2]. Spheroids formed with a single cell type are primarily used to investigate tumor microenvironment regulators and assess responsiveness to therapy, including metabolic and proliferative gradients that can alter the sensitivity of hypoxic tumor cells or their resistance to chemotherapy [3].

However, monocultures may not fully represent the characteristics found in the tumor microenvironment. The tumor microenvironment (TME) plays a significant role in cancer development and therapeutic resistance. It consists of two essential components: a cellular compartment that includes fibroblasts, myofibroblasts, endothelial cells, pericytes, smooth muscle cells, adipocytes, macrophages, lymphocytes, and mast cells [4]; and a non-cellular compartment composed of a network of extracellular matrix (ECM) containing matrix proteins, glycoproteins, glycosaminoglycans, proteoglycans, transforming growth factor-β (TGF-β), vascular endothelial growth factor (VEGF), platelet-derived growth factor (PDGF), and hepatocyte growth factor (HGF) [3,5,6].

Fibroblasts secrete growth factors such as hepatocyte growth factor (HGF), fibroblast growth factor (FGFs), and CXCL12 chemokine, which promote the growth and survival of malignant cells and stimulate the migration of other cells into the TME [7]. Under normal conditions, fibroblasts remain inactive in each organ, but they can be activated in response to signals triggered by the healing process or immune system cells [8]. Normal fibroblasts associated with cancer cells can be transformed into cancer-associated fibroblasts (CAFs) by a process known as fibroblast corruption [9,10].

Studies have shown that the gene expression profile of lung tumor cells changes when cultured with fibroblast-conditioned medium. Shintani et al. demonstrated that lung tumor cells benefit from increased proliferation, invasive ability, and resistance to cisplatin treatment in the presence of fibroblasts [11].

Another crucial cell population within the TME is represented by the macrophages. They are a heterogeneous population of innate myeloid cells originating from their precursor, the monocyte. These cells may undergo differentiation or polarization in the blood or tissues and can assume multiple phenotypes in response to constant changes in the microenvironment [12]. Cancer-associated macrophages (TAMs) are cells that drive the tumor inflammatory response, and are generally associated with a poor prognosis [13]. TAMs are densely organized in hypoxic regions of the tumor, where the same hypoxic microenvironment stimulates cells to overexpress transcription factors such as hypoxia-inducing factor (HIF) or other molecules such as platelet-derived growth factor (PDGF), adrenomedullin (ADM), matrix metalloproteinase (MMPs) and transforming growth factor (TGF-β) [14,15]. Therefore, it has been reported that the polarization of TAMs depends on the TME in which they are located, and this is due to the secretion of soluble and insoluble factors [16]. Co-culture models have shown that the interaction of monocytes with other cell types, specifically non-small cell lung cancer (NSCLC) cells, facilitates macrophage polarization [17,18].

Tumor cells can reside in and transform the stroma, alter the surrounding connective tissue, and modify the metabolism of resident cells, producing a permissive stroma [19]. Several studies have demonstrated the contribution of the tumor stroma in the development and progression of various types of tumors [6]. Immune and inflammatory cells activate the production of chemokines, cytokines, and exosomes, which lead to local tissue remodeling and can contribute to resistance to chemotherapy [3,19].

The study of spheroids in co-culture becomes important, as it is necessary to have a greater context for the interaction that the different cells present in the TEM. Co-culture models prove to be important in the study of tumor physiology, both for the formation of metabolite and chemical gradients, for the creation of a hypoxic environment, and for matrix–cell and cell–cell interactions. Such models can provide valuable insights into the processes that occur in vivo.

This study focuses on the generation of homotypic and heterotypic spheroids using non-small cell lung cancer (NSCLC) cell lines with either fibroblasts (FDH) or macrophages (Mcf), and the characterization of them in terms of generation time, morphological characteristics, cellular conformation, cell viability, cell cycle profile, and migratory capacity. Additionally, the response of both homotypic and heterotypic spheroids to treatment with known chemotherapeutic drugs and compounds with potential chemotherapeutic activity were evaluated. This research provides valuable insights into spheroid behavior and their response to therapeutic interventions.

## 2. Materials and Methods

### 2.1. Cell Lines

The LC-HK2 cell line, a human non-small cell lung carcinoma cell line, which was established in our laboratory [20], the commercial line A549 (non-small cell lung cancer), and THP1 of acute monocytic leukemia were grown in Dulbecco’s modified Eagle’s minimal medium (DMEM) with F-12 nutrients from Sigma, supplemented with 10% fetal bovine serum (Cultilab). The cultures were kept at 37 °C, with the atmosphere containing ~5% CO_2_. The culture medium was changed every two to three days, and the cells were subcultured regularly. Subcultures were obtained by dissociation with 0.05% trypsin and 0.02% EDTA solution.

### 2.2. Primary Culture of Human Fibroblasts (FDH)

A short-term culture of human dermal fibroblasts (FDH) was obtained from discarded breast reduction surgery. Fragments of at least 1 mm of skin were made and placed in bottles by the hanging drop method. After adhesion, they were maintained until the fibroblasts occupied the largest surface of the bottle. Subcultures were obtained by dissociation with a 0.05% trypsin solution and 0.02% EDTA. Discards were obtained from patients (Caucasian women, non-smokers, healthy, between 20 and 45 years old) undergoing mammoplasty surgery in collaboration with Dr. Ricardo Boggio, after approval by the Ethics Committee of the Institute of Biomedical Sciences of the University of São Paulo and after obtaining the free and informed consent term, in accordance with Good Clinical Practices (Resolution n°466/12-Brazil). All the material collected was used in the project, with no discards. CAAE: 84511418.2.0000.5467.

### 2.3. Differentiation of Monocytes into Macrophages

Differentiation of the THP-1 cell line into macrophages was induced, to differentiate into macrophages with the addition of PMA. A total of 25 nM of PMA was added for each 1 × 10^6^ of cells, and after 48 h the medium was changed and differentiation into macrophages was observed.

### 2.4. Generation of Homotypic and Heterotypic Spheroids

Cells grown in a monolayer were subjected to enzymatic dissociation with a solution of 0.2% trypsin +0.02% EDTA. After neutralizing the action of trypsin with culture medium, the cells were counted in a Guava EasyCycle mini flow cytometer (Millipore Biosciences, Temecula, CA, USA) and 1 × 10^4^ cells were added in 96-well plates with agarose bottom (1%). For the generation of heterotypic spheroids, 5 × 10^3^ cells were plated for each cellular type (1:1 proportion). Spheroids were sustained in DMEM-F12 medium supplemented with 10% FBS; the medium was changed every three days, and the plates with the spheroids were kept in the oven under the conditions mentioned above. The images of the spheroids were obtained by the digital inverted light microscope EVOS AME-3302 (AD, Leusden, The Netherlands).

### 2.5. Measurement of the Diameter of the Spheroids

The spheroids were kept in the oven for 7 days, after which the spheroids were photographed with an inverted microscope Evos AME-3302 (AD, Leusden, The Netherlands). The measurement of spheroids was performed with the ImageJ program; 10 spheroids were measured on different days in 3 independent experiments.

### 2.6. Cell Migration Assay

After 7 days in culture, the spheroids were transferred to an adherent surface, and 72 h later, the plates with the migrating cells from the spheroid were photographed, and we proceeded to immunofluorescence reaction, and 5 spheroids were measured in 3 independent experiments using the ImageJ program.

### 2.7. Viability Assay with Hoechst and PI

After treatments, the spheroids were stained with a solution of HO (Hoechst 33342) and PI (propidium iodide) and left in the study for at least 2 h, after which the 3 spheroids each condition were photographed with a Lionheart microscope from Biotek.

### 2.8. Cell Cycle Analysis

After treatments, spheroid cells were resuspended using trypsin-EDTA, centrifuged at 1200 rpm for 10 min, washed with phosphate buffered saline A (PBSA, without Ca^+^ and Mg^+^) and centrifuged. Samples were fixed with 75% methanol at 4 °C for 1h and washed with PBSA. The DNA was stained with PI (10 μg/mL) and treated with RNase (10 µg/mL) at 4 °C for 1 h and quantified by flow cytometer (GUAVA EasyCyte Plus, Hayward, CA, USA). The assay was conducted three times in replicates, and the results were expressed as mean ± SD of the percentage of cell distribution in each cell cycle phase (G1, S, and G2/M).

### 2.9. Immunofluorescence and Fluorescence Assay

For immunostaining, samples were fixed with 3.7% formaldehyde (Sigma-Aldrich, HAVERHILL, MA, USA) for 30 min, permeabilized with Triton X-100 (0.5%) for 30 min and incubated overnight with primary monoclonal antibodies and for 2 h with secondary antibodies. Nuclei were labeled with DAPI (1:100) or propidium iodide (10 mg/mL) from Sigma-Aldrich (St. Louis, MO, USA). The slides were mounted with Vecta-Shield from Vector Laboratories (Newark, CA, USA) and analyzed by laser scanning confocal microscopy from Leica TCS SP8 (HE, DE) or with fluorescent microscopy Lionheart from Biotek.

### 2.10. Transmission Electron Microscopy (TEM)

Spheroids were fixed for 2 h with 2.5% glutaraldehyde and 2% formaldehyde in 0.1 M sodium cacodylate buffer, pH 7.2. The fixed samples were washed in 0.1 M sodium cacodylate buffer, pH 7.2, and post-fixed in 1% osmium tetroxide. The tissues were dehydrated in a graded ethanol and propylene oxide series. Resin infiltration was done with a 1:1 mixture of propylene oxide and EPON (Electron Microscopy Science, Hatfield, PA, USA) for 5 h, followed by pure Epon for 5 h. Next, the material was embedded in Epon and polymerized for 48 h. Semi-thin sections were cut using an ultra-microtome and stained with toluidine blue. Appropriate regions of the spheroids were then thin sectioned at 70 and 90 nm and stained with 4% uranyl acetate and a 10% lead citrate solution. The material was analyzed with a Jeol 1010 transmission electron microscope at 80 kV.

### 2.11. CellTiter-Glo 3D Cell Viability Assay

After of treatment for 48h with Staurosporine 10 µM, Doxorubicin (10, 4 and 2 µM), CDDP (cisplatin 200, 100 and 50 µM) and CA5 (50, 25 and 5 nM). The viability of the spheroids was assessed by ATP quantification utilizing the CellTiter-Glo 3D Cell Viability kit (Promega, Madison, WI, USA) according to the manufacturer’s instructions. The measurement represents the average of 4 spheroids per condition in 3 independent experiments.

### 2.12. Statistical Analysis

Results were expressed as means (SD). Data were submitted to the ANOVA test, followed by Dunnet’s test for multiple comparisons with the control. *p*-value < 0.05 was considered statistically signific.

## 3. Results

### 3.1. Homotypic and Heterotypic Spheroid Generation

The spheroids were generated by the liquid overlay technique under non-adherent conditions, and for this purpose 1% agarose was added to the bottom of the plates. In homotypic spheroids, LC-HK2 and A549 tumor cells aggregate on the second day after plating, unlike heterotypic spheroids with FDH that form spheroids on the first day after plating. The proportion of generation for the heterotypic spheroids was 1:1; this proportion was optimal for both heterotypic spheroids.

The heterotypic spheroids with THP-1 (LC-HK2/Mcf) are formed after 48 h, the time corresponding to the differentiation of monocytes to macrophages. After a few days in culture, the spheroid cells became more cohesive, and after 7 days of culture, the spheroids, both homotypic and heterotypic, presented a more compact center when observed under a light microscope (Figure 1A). The diameter of the spheroids on different days of cellular culture was measured. The A549 homotypic spheroids showed a decrease in size, on the last day (day 7); they measured 420 µm (Figure 1B). On the other hand, the heterotypic A549/FDH spheroids started with a smaller diameter compared with A549 spheroids, and after 7 days in culture the diameter was ~330 µm.

The LC-HK2 homotypic spheroids also decreased in size; after 7 days the diameter was ~400 µm, and the first day it was ~600 µm (more details in the figure). The LC-HK2/FDH heterotypic spheroids started with a smaller diameter than the homotypic, and after 7 days the diameter achieved ~320 µm, but they seemed to have a less variable diameter than the A549/FDH spheroids. The heterotypic LC-HK2/Mcf spheroids also showed a decrease in diameter, and after 7 days the diameter was ~360 µm (Figure 1B).

The homotypic spheroids formed only with FDH had the smallest diameter of all (Figure 1B), considering that the plating for the generation of the spheroids maintained a homogeneous density for all spheroids. So, after 7 days in cellular culture they achieved the final diameter of ~260 µm.

The number of cells in spheroids on different days in cellular culture was quantified. It was observed that the homotypic spheroids LC-HK2 and A549 (Figure 1C) showed an increase in the number of cells that make up each spheroid, and this variation in the number of cells was greater in the A549 spheroids. On the other hand, the heterotypic spheroids showed a decrease in the number of initial cells with a more pronounced effect on the A549/FDH spheroids, whereas the LC-HK2/Mcf spheroids on the last day analyzed showed an increase in the number of cells (Figure 1C). This variation in the number of cells follows the variation in the size of heterotypic spheroids with FDH, but the relationship is not direct in other types of spheroids.

### 3.2. Variation in the Diameter of Homotypic Spheroids

Some factors were observed that may vary in the spheroids, such as the initial density, the well area in which they are cultured, and even the cell lineage. The density of cells in the A549 cell lines affects the diameter of the spheroids. When spheroids were generated with a lower cell density (1 × 10^3^), their diameter increased in the first few days (as shown in Figure 2A). After 7 days, the diameter was around 370 µm, and after 10 days it was around 380 µm. No further diameter variations were observed in the later days.

The diameter of the spheroids may be constrained by various factors, including the size of the 96-well plate used for their generation. To mitigate this issue, the spheroids were initially cultured in a 96-well plate and subsequently transferred to a 24-well plate after five days. The spheroids of the A549 lineage after a few days have a darkened center, and the cells seem to be detaching from the spheroid (Figure 2B).

On the other hand, the LC-HK2 spheroids did not present a heightened darkened center, and, unlike the A549 spheroids, the LC-HK2 cells remained cohesive in the spheroids; after approximately 30 days in cell culture, the spheroids appeared larger (Figure 2C). Thus, when measuring the diameter of the A549 spheroids, the variation observed seemed to be the consequence of the greater number of detached cells from the spheroids. The diameter of the LC-HK2 spheroids increased over the days, achieving ~670 µm on the last day analyzed (Figure 2C).

### 3.3. Characterization of Cell Population of Homotypic and Heterotypic Spheroids

The homotypic spheroids of the A549 and LC-HK2 lines exhibited mitotic cells mainly in the periphery (Figure 3). This was also observed in the heterotypic spheroid LC-HK2/FDH. In the FDH homotypic spheroid, mitoses were not observed, and the elongated fibroblasts forming the spheroid could be seen. These cells maintained their elongated morphology in the spheroid. A549, LC-HK2, and THP-1 cells exhibited different morphologies when in the spheroids, presenting rounded morphologies.

As observed by the fluorescent staining of the actin cytoskeleton (red) and the immunofluorescence of the microtubules, the A549 cells that form the spheroids do not appear to show very strong aggregation among themselves, appearing to be a less cohesive structure than the cells that form the LC-HK2 spheroids. These spheroids showed greater aggregation between the cells, and this was also very similar in the cells forming the LC-HK2/Mcf spheroids.

Cytokeratin 18 is an intermediate filament protein highly expressed in carcinoma cells. Therefore, the immunofluorescence of cytokeratin 18 was performed to differentiate the NSCLC from the FDH in the heterotypic spheroid formed. Distinct distribution of A549 cells (Figure 4A’ in green) were observed in the A549/FDH heterotypic spheroids, as shown in Figure 4A, compared to the more central localization of LC-HK2 cells (Figure 4B’ in green) in the LC-HK2/FDH heterotypic spheroids depicted in Figure 4B. It can also be observed that the heterotypic spheroids seem to remain a much more cohesive structure than their homotypic counterparts, which would reveal the effect of FDH on the generation of heterotypic spheroids.

Semi-thin sections obtained from the preparations for transmission electron microscopy of homotypic and heterotypic spheroids are showed in Figure 4C. The A549 spheroid cells were observed to have rounded morphology and appear looser, due to smaller cell contact. The LC-HK2 spheroid cells also displayed a rounded morphology, resulting in a more cohesive structure.

The A549/FDH heterotypic spheroid cells were observed to be in closer spatial arrangement than the homotypic A549, resulting in a more cohesive spheroid structure. Similarly, the LC-HK2/FDH heterotypic spheroid cells also appeared to form more cohesive spheroids. These observations suggest that FDH plays a crucial role in spheroid conformation by promoting a cohesive morphology that is distinct from the less cohesive morphology found in A549 homotypic spheroids. Lastly, the cells of the LC-HK2/Mcf heterotypic spheroid were found to form a less cohesive structure compared to the heterotypic spheroids containing FDH.

Transmission electron microscopy (TEM) was used to examine the cellular organization of both homotypic and heterotypic spheroids. In general, the spheroids of A549 and LC-HK2 show a typical structure of cells active metabolically (Figure 5), exhibiting nuclei with nuclear pores, euchromatin, and condensed heterochromatin. Many prominent nucleoli were observed, and in the cytoplasm we observed numerous mitochondria; the endoplasmic reticulum with ribosomes distributed throughout the cell, surfactant substance-containing vesicles, lysosomes, and the Golgi apparatus, characterized by the shape of cisterns, were also observed. Additionally, multilamellar bodies in whorl structures were present. Cellular processes such as apoptosis and autophagy were also observed.

While cell junctions play a crucial role in cell adhesion and communication, they may not always be visible. However, cell interdigitations, which refer to the overlapping of cell membranes and protrusions into each other’s spaces, can be observed under certain imaging techniques, such as transmission electron microscopy. Therefore, while cell junctions may not be observable, the presence of interdigitations between cells can still provide important information about cell-to-cell interaction and tissue architecture.

In the A549/FDH heterotypic spheroids (Figure 6), nuclei with euchromatin and condensed heterochromatin were observed. Some elongated nuclei were also noted, which may be characteristic of fibroblast nuclei due to their arrangement, whereas the nuclei of A549 cells appeared more rounded. We can observe very evident nucleoli, a great number of mitochondria, endoplasmic reticulum distributed in several places of the cell, cellular interdigitations and cellular processes like autophagy.

The LC-HK2/FDH heterotypic spheroids (Figure 6) presented nuclei with euchromatin and heterochromatin, evident nucleolus, large numbers of mitochondria, vesicles containing surfactant substances, and very dense cytoplasm containing intermediate filaments, along with the Golgi complex and endoplasmic reticulum.

The LC-HK2/Mcf heterotypic spheroids (Figure 7) showed nuclei with chromatin and condensed heterochromatin, evident nucleolus, large numbers of mitochondria, endoplasmic reticulum containing ribosomes, and double membranes in the cytoplasm close to the nucleus; these double membranes may be evidence of mitochondrial rearrangement. In what appear to be cells that are more at the periphery, we can see cells with different characteristics, with more electron-dense organelles, and with what appear to be thicker cell extensions but which are similar to lamellipodia, so these cells could be macrophages.

### 3.4. Viability and Cell Cycle of Spheroids

Homotypic and heterotypic spheroids were stained with Hoechst 33342 (blue) and propidium iodide (red) to observe their cell viability after 7 days in culture (Figure 8A); propidium iodide is able to penetrate the membrane of damaged cells that are being routed to death. The spheroids were stained with a solution containing both dyes for ~1 h and 30 min at 37 °C.

A small number of red-stained cells were observed in the spheroids. In most spheroids, these red-stained cells were located at the center, while in some cases they were more distributed. In homotypic spheroids, the red-stained cells were observed to be less distributed at the center, whereas in heterotypic spheroids with fibroblasts they were observed to be more distributed away from the center. In the heterotypic spheroids LC-HK2/Mcf, the cells stained in red display a distribution that is not limited to the center of the spheroids. This is evident from the observation that the distribution is present in various locations throughout the spheroid.

The cell cycle of homotypic and heterotypic spheroids after 7 days in culture was analyzed. It was observed that all spheroids had most of the cell population in the G1 phase, with values between 71 and 85%. The lowest value for the homotypic spheroids is A549 (71.5%), with the highest values for the heterotypic A549/FDH (83.6%) and LC-HK2/FDH (84.84%). The A549/FDH spheroids showed two peaks in the G1 phase, which is probably the result of the difference in the DNA content of both populations. The population in the G1 phase was greater than that found in the homotypic spheroids of LC-HK2.

The S phase cell population was smaller for homotypic and heterotypic spheroids, with the homotypic spheroids A549 maintaining the highest percentage, at 12.69%, and the heterotypic spheroids LC-HK2/Mcf presenting the lowest percentage, at 3.81%. On the other hand, the cell population of the G2/M phase presented different percentages in the spheroid cells; the homotypic A549 spheroids presented 15.79%, and the LC-HK2 in turn obtained 11.96%. Heterotypic spheroids with FDH showed lower percentages, with 8.96% in A549/FDH and 9.78% in LC-HK2/FDH. The heterotypic spheroids LC-HK2/Mcf presented 15.74% of their population in the G2/M phase.

### 3.5. Migration of Spheroid Cells

The spheroids were placed on an adherent surface for the cell migration. It was observed that the LC-HK2, A549, LC-HK2/FDH, A549/FDH, and LC-HK2/Mcf spheroids adhered in the first 24 h, but the FDH spheroids took much longer to adhere to the surface. After 72 h (Figure 9A,B), the measurement of the area occupied by migrating cells from the spheroids was performed. It was observed that the A549/FDH heterotypic spheroid area of migration was less than the area of A549 cells of homotypic spheroids, but that this area was similar to the area occupied by the cells of FDH spheroid.

The migration area of LC-HK2 and LC-HK2/FDH spheroids appeared similar (Figure 9A,B), despite being smaller than that of the homotypic spheroids. The LC-HK2/Mcf heterotypic spheroids had a smaller migration area than the other LC-HK2 spheroids, but the area was larger than that of FDH spheroids.

LC-HK2 and LC-HK2/FDH spheroids with 7 days in culture are shown in Figure 9C. Cells with increased cytoplasm can be observed on the surface of these spheroids (highlighted), indicating a potential change in the surface cells. The work of Stadler and colleagues isolated these cells and observed that they showed advantages in migration and invasion.

Immunofluorescence of migrated cells from LC-HK2 and LC-HK2/FDH spheroids (Figure 9D) showed some morphological modifications, with very large multinucleated cells; however, aberrant cell divisions were not observed. In Figure 9A, we can also observe that the cells that migrated from the LC-HK2/Mcf spheroids showed an altered morphology, being more elongated and presenting very different extensions.

### 3.6. Response of Chemotherapy Compounds and Chemotherapy Potentials of Heterotypic Spheroids LC-HK2/FDH and LC-HK2/Mcf

To analyze the responses of spheroids to different chemotherapy compounds and chemotherapy potentials, treatment of LC-HK2 and macrophages or fibroblasts heterotypic spheroids was performed. For this, doxorubicin (Dox), cisplatin (CDDP), chromomycin A5 (CA5) and Staurosporine (STS) were added to spheroid cultures for 48 h of treatment. The viability assays were carried out by staining with Hoechst and IP, measuring the diameter of the spheroids, and finally measuring the production of ATP.

An important feature of spheroids is the penetration process, which can become a more difficult process due to the architecture of the spheroid. Thus, Dox presents autofluorescence, which helps us to check its penetration into spheroids. The LC-HK2 spheroids (Figure 10A) showed that the higher concentration of Dox had an impact on cell viability, which can also be evidenced by the decrease in the diameter of the treated spheroids (Figure 10B). In heterotypic spheroids, there is no great difference in the diameter of spheroids treated with Dox, but one can also observe a more intense red center, which may also indicate an accumulation of Dox in the center of the spheroid. Another characteristic that must be considered is the difference in diameter between homotypic and heterotypic spheroids. Heterotypic spheroids are slightly smaller than homotypic spheroids; nevertheless, the response for homotypic and heterotypic with FDH is similar. However, the heterotypic with macrophages exhibited lower viability.

The spheroids treated with cisplatin at different concentrations did not seem to show an increase in the number of cells stained with IP in any of the three types of spheroids, nor was the diameter significantly altered (Figure 10A,B). Nonetheless, spheroids treated with different concentrations of CA5 showed a decrease in the diameter and integrity of the spheroid depending on the dose used. Additionally, a greater distribution of cells stained with PI was observed. Proportionally to the size, it was inferred that the homotypic spheroids, in relation to the heterotypic ones, presented a greater loss of integrity or size when treated with CA5 (Figure 10A,B).

The measurement of ATP production has an indirect relationship with cell viability, which can be seen in the spheroids treated with Dox, CDDP, CA5, and Sts (Figure 10C). Sts was used as a control for cell death, and, as can be seen in Figure 10B, there is an increase in the diameter of the LC-HK2 and LC-HK2/FDH spheroids, which is associated with the loss of cohesion between cells as a result of the death. In relation to homotypic and heterotypic spheroids with fibroblasts, it was observed that Dox, CA5, and CDDP affected the cell viability of LC-HK2/Mcf spheroids in a more dramatic way, by measuring ATP production. This is probably because the compounds can penetrate these spheroids more efficiently.

In LC-HK2/FDH and LC-HK2 spheroids, it was observed that CA5 decreases ATP production. This is better observed in LC-HK2 spheroids, which can be explained because LC-HK2/FDH spheroids are more cohesive structures than their homotypic counterparts. However, with regard mainly to CDDP, we can see that this measure could not be associated with cell viability, but rather with the possible metabolic stress that this drug is inducing in these spheroids.

## 4. Discussion

Currently, one of our challenges is developing cell culture models that reflect the conditions found in vivo. This way, we can develop tools that improve the study of drugs and increase their effectiveness against cancer.

Uniform spheroids can be formed by using microwells that prevent cells from adhering to the wells and aggregating between them [21]. Then, in order to achieve less variability in the generation and size of the spheroids, we can use the liquid overlay method in microwells with an agarose bottom, thus preventing the adhesion of the spheroids to the bottom of the plate. We were able to generate both homotypic and heterotypic spheroids, with diameters showing relatively minor variation. It is noteworthy that the dermal fibroblast spheroids exhibited an even smaller degree of diameter variation. This finding could be attributed to the properties of dermal fibroblasts, which are known to play a crucial role in wound healing and tissue regeneration. The ability to form spheroids with a consistent diameter is crucial for the accurate evaluation of various cellular parameters, such as viability, proliferation, and migration.

Tumor cells can survive in unfavorable conditions such as deprivation of nutrients, oxygen, or growth factors, inhibition by contact with neighboring cells, or loss of anchorage in the cell layer [6]. Cell adhesion, aggregation, and growth in 3D cultures can be artificially produced by adding extracellular homologs such as collagen or matrigel. Other methods for generating 3D cultures include liquid overlay, fiber mesh made of biocompatible polymers, solid or porous granules, and extracellular matrices and their substitutes. These methods also require the addition of artificially produced supplements. The search for mimicking the ECM in the tumor microenvironment has led to the emergence of different biomaterials, as well as the addition of ECM components [22,23].

This study sheds light on the contribution of fibroblasts to the complexity of spheroids, as evidenced by cytoplasmic changes observed through transmission electron microscopy of LC-HK2/FDH spheroids. Interestingly, the incorporation of FDH into the spheroid composition was found to expedite the aggregation of the constituent cells, which differs from the aggregation time observed in homotypic spheroids. These findings suggest that FDH association could be leveraged to enhance the efficiency of spheroid generation assays for investigating cellular behavior in 3D microenvironments.

Fibroblasts are responsible for synthesizing and organizing ECM components, generating biological signals, growth factors, angiogenic factors, and remodeling enzymes such as matrix metalloproteinase (MMP) [24]. The composition and physical properties of the ECM influence the functional behavior, cell growth, morphology, and survival of the cell population. Changes in these properties play a key role in tumor development, progression, and metastasis [25,26].

In cancer, fibroblasts remain activated, and there is a failure to remove them by apoptosis, with them thus becoming important contributors to tumorigenesis [19]. It is currently believed that most active fibroblasts and CAFs are derived from resident fibroblasts that have transdifferentiated in response to factors secreted by tumors such as TGF-β [27]. The study by Fromigué and colleagues demonstrated that co-culture between A549 cells (NSCLC) and normal fibroblasts induce the expression of ST3, a protein expressed by invading fibroblasts. They also observed an increase in the expression of MMPs [24].

In addition to playing an important role in ECM deposition, active fibroblasts are responsible for the formation of a capsule that surrounds the tumor. This capsule is associated with a lower capacity for tumor cell invasion, but, at the same time, it makes it more difficult for the chemotherapy drugs to penetrate [4,28]. When generating heterotypic spheroids with FDH, it was noticed that the cell resuspension process of the spheroids with trypsin took longer than that of the homotypic spheroids and heterotypic spheroids with macrophages. Thus, we can infer that in this case, the fibroblasts were responsible for a greater deposition of ECM, an issue that can also be observed in the penetration of drugs, since the LC-HK2/FDH spheroids had the lowest decrease in cell viability.

Unlike findings from other studies involving NSCLC and fibroblasts, we were able to form spheroids from the A549 cell line. Additionally, there were differences observed in the morphology of A549 cells and fibroblasts, with the latter having an elongated morphology, as seen in TEM. Curiously, when LC-HK2 cells and short-lived fibroblasts (FDH) were combined to form heterotypic spheroids, no discernible morphological difference between the two cell types was observed. However, this contrast in morphology had already been reported in a previous study involving Colo 699 cell lines (NSCLC) and SV80 fibroblasts (immortalized cell lines). In this same study, the production of ECM was demonstrated and associated with a more rounded morphology, as well as a more homogeneous surface of the spheroid when fibroblasts were added [23].

The compaction of the spheroids can be attributed to both cellular death and an increase in the production of ECM components, which contribute to enhanced cell cohesion. The phenomenon of cellular death in this case is a result of the anoikis process. It is noteworthy that not all tumor cells are able to withstand anoikis [6]. It is interesting to observe that FDH cells, despite experiencing cellular death, were able to form tightly packed spheroids.

Spheroids exhibit crucial characteristics, such as a proliferative and metabolic gradient, which impact pharmacological efficacy. The necrotic center in spheroids is formed due to restricted access to nutrients and oxygen, leading to low pH and the presence of metabolites. Cells in the center adapt and become quiescent. Distinct zones were formed within spheroids, with proliferative cells on the outer edges and quiescent and necrotic cells in the center, similar to solid tumors. Spheroids, without an external matrix, secrete extracellular matrix to create a tissue-like microenvironment, resulting in hypoxia [29,30].

In the tumor, only a small subset of cells performs an action, while most become dormant or die. These solitary cells are known as dormant tumor cells (DTCs), and they are characterized by their lack of proliferation and death while they remain silent [3]. In the spheroids generated, a center of dead cells was observed, which can be seen in the viability assay by staining with HO and IP. In addition, a population of quiescent cells was observed, characterized by a high percentage of cells in the G1 phase and a decrease in the S phase in all spheroids. The spheroids studied had all the expected cell populations, including proliferative cells on the outer edge.

In both tumors and organs, the average interstitial oxygen decreases as one moves away from the blood vessels, and different degrees of hypoxia are observed, ranging from mild to severe. This is due to the limited oxygen diffusion in tissue, which depends on the rate of tissue oxygen consumption and is restricted to a tissue thickness of approximately 130 µm [4]. Spheroids with an average diameter size between 150 and 300 µm show an intermediate normoxic quiescent zone, and a hypoxic zone in the center [29,31]. As observed, the spheroids generated had a final diameter between 300 and 400 µm after 7 days in culture, which could indicate zonation within the spheroids.

Hypoxia induces cellular processes reprogramming through hypoxia-inducible factors (HIFs), a family of transcription factors. These HIFs mediate primary adaptive responses to changes in oxygen levels in the environment [4]. Hypoxia is a strong stimulus in the tumor microenvironment and leads to the activation of autophagy. Cancer cells employ autophagy as a mechanism to support their survival and preserve cell integrity [32]. The homotypic and heterotypic spheroids also showed cellular processes such as apoptosis and autophagy.

Autophagy is the lysosomal degradation process that packages damaged proteins and organelles into double-membrane vesicles and transports them to the lysosome for degradation [33]. It is involved in several pathologies and has a dual role in tumorigenesis, both promoting cell progression and resistance to chemotherapy [34].

Studies of NSCLC have shown that an acidic pH can induce autophagy flux, promoting lung cancer cell survival in vivo and in vitro. This pH can also cause ROS production, resulting in endoplasmic reticulum stress [33]. The acidic tumor microenvironment contributes to drug resistance by creating a physiological barrier to weakly basic chemotherapy drugs, thereby preventing their access to cancer cells [35,36].

In the homotypic and heterotypic spheroids generated, lamellar bodies were observed. Lamellar bodies are organelles associated with lysosomes to produce surfactant substances, and were also associated with autophagosomes in mouse lung pneumocytes [37,38]. These lamellar structures can be present in normal cells, but also in cells that have suffered some type of stress or in pathological conditions [39].

Many mitochondria were observed, as well as a variety of morphology and size in both homotypic and heterotypic spheroids. The shape, size, and number of mitochondria are controlled by the dynamics of two opposing processes, fission and fusion, and mitochondrial fission is predominant in tumor cells. Mitochondrial fragmentation is a result of excessive fission [40]. Mitochondria form a network that contacts the nucleus, endoplasmic reticulum, Golgi complex, and cytoskeleton. Mitochondrial morphology changes based on cellular demands, and can include branching, bending, retractions, and changes in the shape and number of cristae [41].

Cancer mitochondria have heterogeneous ultrastructure, with changes including outer membrane buckling, crystal disorganization, matrix myelin figures, vacuoles, and distorted shapes. Despite these changes, cancer mitochondria do not differ much from normal mitochondria in function, substrate oxidation rate, phosphate ion transport, passive swelling, and other factors [42]. In tumors such as pancreatic or salivary duct carcinoma and others, many mitochondria have been observed. Mitochondria exhibit variability and abnormalities in number, size, and shape, even within the same specimen, as well as a degree of severity of internal ultrastructural changes [40]. Moreover, in heterotypic spheroids with macrophages, the presence of double membranes can be observed, which could be an indicator of mitochondrial rearrangement.

Migrating cells survive in the circulation, with their extravasation into a nearby or distant organ/tissue subsequently colonizing a secondary site [3]. For invasion and metastasis, the interaction between fibroblasts, immune cells, angiogenic cells, and their factors is essential [19]. Unlike other studies with spheroids, where the size and invasiveness of cells reflect their migratory capacity [29], the migration of cells on an adherent surface in LC-HK2 spheroids did not depend on their size. Heterotypic spheroids (LC-HK2/FDH), which had a slightly smaller final diameter than homotypic spheroids (LC-HK2), showed a migration area that was very similar to each other. In contrast, the LC-HK2/Mcf heterotypic spheroids had a smaller migration area than the LC-HK2 spheroids. Previous work on cancer pancreatic cells demonstrated that cells selected for resistance to anoikis had higher rates of migration and invasion than cells that were not selected [43].

Compared to A549 spheroids, those formed with A549/FDH cells exhibited a smaller migration area, indicating a potential influence of fibroblasts on the migratory response of A549 cells. These findings highlight the impact of spheroid composition on cellular processes, and demonstrate how they can differ depending on the cell type. Specifically, our results emphasize the role of fibroblasts in regulating cell migration and highlight the need for further investigation into complex interactions between different cell types in 3D microenvironments.

The spheroids generated in this study exhibited both common and individual characteristics that are interconnected and subject to further study. The importance of a 3D system that allows for the observation of interactions between different cell types and the creation of necessary characteristics is highlighted. Co-culture models, where different cell types are grown in the same culture dish, provide a greater degree of in vivo similarity compared to monocultures and can be used to study cell–cell interactions ex vivo. However, analyzing cellular interactions becomes challenging, since these interactions change depending on the environment, and others occur within a defined pH range [22,30]. Three-dimensional models, such as spheroids, generally reflect cellular behavior in tissues and recapitulate cellular heterogeneity in tumors. The highly compact spheroids formed in this study indicate cell–cell and cell–matrix contact [21,44].

LC-HK2/FDH spheroids exhibited enhanced resistance to drug penetration compared to other generated spheroids, even when higher concentrations were used, surpassing those typically employed in monolayer studies. The configuration of spheroids can significantly influence drug delivery and efficacy outcomes. Therefore, it is essential to thoroughly characterize spheroids to minimize variability across experimental setups, enabling a better result comparison and reproducibility.

As observed, the concentration of the compounds used in the spheroids was higher than the typically employed IC50 concentration in monolayer cultures. It is also known that the gene expression profiles of the 3D culture have been shown to more accurately reflect the expression profiles observed in the 2D culture [31]. The lack of similarity in architecture between monolayer culture and in vivo conditions can lead to erroneous conclusions. Some assays and drugs that exhibit good activity against cancer cells in monolayer culture may not have the same response when translated to clinical trials [30,44]. Therefore, it is crucial to include 3D studies in the screening of compounds to support the findings of monolayer studies [30]. It is important to consider that conducting multiple feasibility tests is necessary to demonstrate, with greater accuracy, the effects of drugs on spheroids. Staurosporine, which was used as cell death control in this study at a high concentration, is a drug isolated from *Streptomyces staurosporeus* (now Lentzea albida). Initially, it was characterized as an exceptionally powerful inhibitor of protein kinase C, displaying remarkable potency. However, further studies have revealed that the compound exhibits a high degree of promiscuity, interacting with numerous other kinases in addition to its primary target [45].

Cisplatin induces cell death in various cell lines through multiple mechanisms. It triggers oxidative stress by generating reactive oxygen species and causing lipid peroxidation. Additionally, cisplatin activates p53 signaling, leading to cell cycle arrest. It down-regulates proto-oncogenes and anti-apoptotic proteins while activating both intrinsic and extrinsic pathways of apoptosis. These combined effects contribute to the cytotoxicity of cisplatin and its ability to eliminate cancer cells [46,47]. Cisplatin is a well-known chemotherapeutic widely used in NSCLC, with an IC50 of 22.5 µM in the A549 cell monolayer [48].

Doxorubicin, like cisplatin, is used in the treatment of NSCLC; it is characterized as a DNA damaging agent, and in A549 cells it has an IC50 of 1.48 µM [49]. Doxorubicin was extracted from *Streptomyces peucetius*. The antitumor activity of doxorubicin (Dox) has been attributed to its ability to intercalate DNA and its association with proteins involved in DNA replication and transcription. Dox is classified as a topoisomerase II poison [50]. Another mechanism of action of Dox may be its ability to generate free radicals that induce damage to cell membranes and DNA [51]. The high therapeutic effects and clinical applications of doxorubicin are compromised due to its high hydrophilicity, short half-life, and the requirement of high doses for treatment effectiveness resulting in cardiotoxicity, extravasation, nephrotoxicity, and myelosuppression [52].

Chromomycin A5, in turn, is a glycosylated tricyclic polyketide, a member of the aureole acid group of antitumor antibiotics. Studies suggest that chromomycin A5 is associated with the transcription factor T-box 2 (TBX2), and could be associated with its anti-proliferative and potential anti-metastatic effects [53]. CA5 would also act as an inducer of ICD (immunogenic cell death) [54]. Chromomycins are highly cytotoxic, and can induce autophagy in melanoma cells [55]. In the monolayer of melanoma, breast cancer, and rhabdomyosarcoma cell lines, CA5 presents an IC50 between 0.8 and 6.5 nM [53]. In this study, CA5 was the compound that had the most pronounced effect on the spheroid cell viability. This effect was observed through a decrease spheroid size and a reduction in ATP production.

It was observed that cisplatin increases ATP production, indicating changes in metabolic rates, as observed in other studies, but this is not associated with cell viability. Something similar can be observed with doxorubicin, but in an inverse way. In the stained result obtained with HO and IP, a center appears much more stained in red, but it does not reflect what happens with ATP production. When analyzing the response to compounds, it was observed that fibroblasts can not only provide a physical barrier, but also influence the metabolic response. In the case of macrophages, they may interact not as physical barriers, but likely through the production of signals that enable the action of compounds on cell viability. However, further studies on the role of these macrophages are necessary. Homotypic spheroids had a larger diameter than heterotypic spheroids. This difference in size suggested that the penetration of compounds into the spheroids could be different. However, despite the variation in diameter, heterotypic spheroids with fibroblasts responded to the tested compounds similarly to homotypic spheroids in terms of ATP production. It is important to consider the impact of spheroid size when analyzing compound responses.

The observation indicates that cisplatin increased the ATP production, suggesting changes in metabolic rates, which align with findings from other studies. However, this increase in ATP production does not align with cell viability. Similarly, doxorubicin showed a similar pattern, but in an opposite manner. In the viability reactions with HO and IP, the central region of the spheroids was more red-stained. Nevertheless, the staining pattern does not correspond to the observed ATP production or cell viability. It could be just doxorubicin accumulation.

In the analysis of compound responses, it was observed that fibroblasts not only act as physical barriers, but also play a role in the production of signals that facilitate the action of compounds on cell viability. These observations highlight the complexity of the interaction between different cell types and compounds in the experimental setup.

The response to treatments may also be modulated by cell–cell interaction, including the interaction between two different cell types, which can influence the process. Much of our knowledge about how ECM can modulate drug responses and contribute to resistance comes from studies of cancer cell interactions with tumor stroma [5,6]. At this stage, the significance lies in generating not only homotypic spheroids, but also heterotypic spheroids composed of stromal cells, along with tumor cells. This combination reflects the interplay between different cell types within the tumor microenvironment and enhances the relevance of studying tumor relapses.

## 5. Overcoming Challenges

In this study, we recognized certain limitations in assessing the production of extracellular matrix components and cytokines in homotypic and heterotypic spheroids. To address this, we advocate for the incorporation of more molecular tools to gain a deeper understanding of spheroid behavior and its impact on tumor progression. Additionally, external factors like hypoxia and nutrient gradients were found to influence matrix and cytokine production within spheroids. Understanding these microenvironmental influences is crucial for unraveling tumor complexities and guiding targeted therapies.

Furthermore, we emphasize the importance of validating our findings in diverse tumor types to ensure the broader applicability of our research outcomes and their potential translational applications in human cancer treatment. In conclusion, by addressing these challenges and pursuing innovative approaches, we can make significant strides in cancer research using spheroids, offering hope for more effective and personalized treatments for cancer patients.

## 6. Conclusions

Homotypic spheroids of NSCLC (A549 and LC-HK2) and heterotypic spheroids of NSCLC with fibroblasts or macrophages were generated. Heterotypic spheroids with fibroblasts form in a shorter time, after the first day, unlike homotypic spheroids, which are generated after two days. Different cell populations corresponding to spheroid zoning sites were characterized, such as dividing cells observed on the spheroid surface, dead and quiescent cells in the center, organelles and cellular processes corresponding to hypoxic zones, or the presence of a diffusion gradient. The homotypic spheroids exhibited a larger diameter compared to the heterotypic spheroids, suggesting that compound penetration would be different. Interestingly, it was observed that heterotypic spheroids with fibroblasts responded to the tested compounds similarly to the homotypic spheroids in terms of ATP production. Additionally, as expected, the concentration of compounds was higher than that found in a monolayer. It was also observed that, to analyze the cellular viability of spheroids, more than one methodology should be used, as the response may vary depending on the compound analyzed, whether homotypic or heterotypic spheroids. Therefore, the complexity of the spheroid should be considered when analyzing the response of compounds in spheroids. Thus, the importance of introducing cells that are part of the tumor stroma is notorious, since the response to different compounds will depend on the complexity of the spheroid; this can affect the results in the screening of new drugs, because it depends on the action they develop.

## Figures and Tables

**Figure 1 cells-12-02790-f001:**
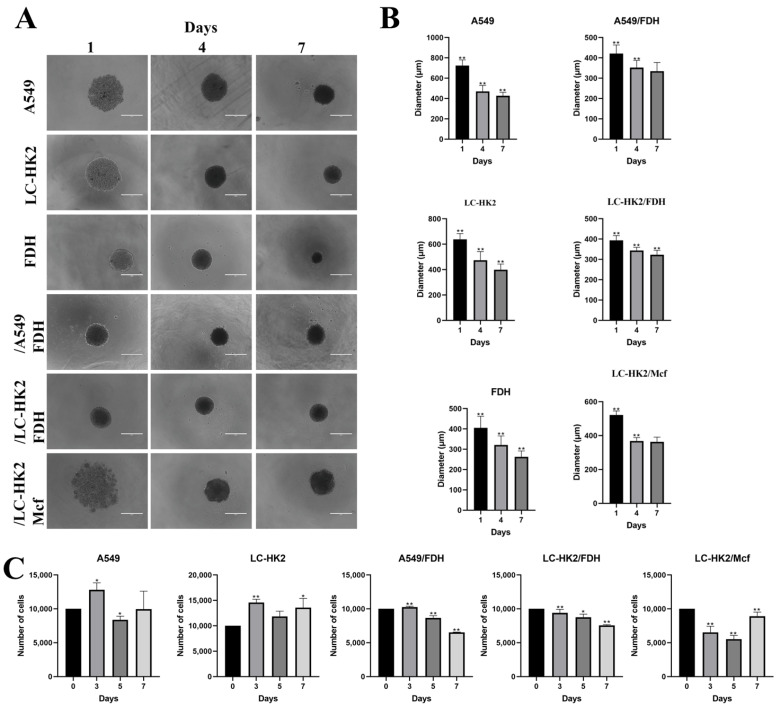
Homotypic and heterotypic spheroid generation. (**A**) Homotypic and heterotypic spheroids on different days in cell culture. Scale bar 400 µm. (**B**) Measurement of spheroid diameters. (**C**) Number of cells/spheroid on different days in culture. Data are shown as mean relative to control in three independent experiments (SD). Statistical analysis was performed using Dunnett’s test with an ANOVA test for multiple comparisons vs. control. * *p* value < 0.05 and ** *p* value < 0.001.

**Figure 2 cells-12-02790-f002:**
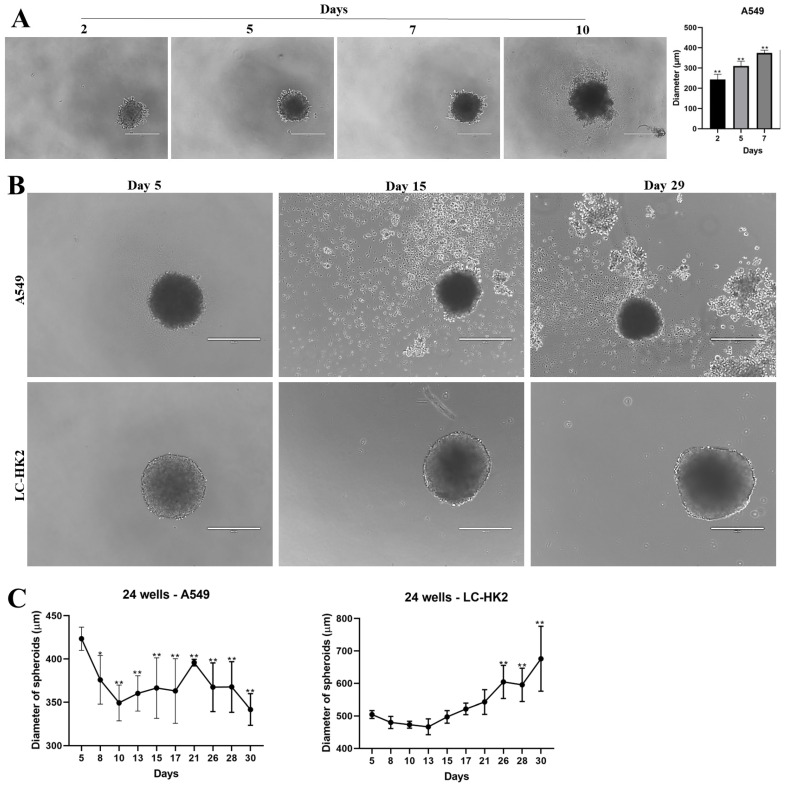
Variation in the diameter of spheroids by density and the size of well. (**A**) homotypic spheroids of the A549 lineage formed with a density of 1000 cells per well. Scale bar 400 µm. (**B**) Measurement of the diameter of A549 and LC-HK2 homotypic spheroids. Scale bar 400 µm. (**C**) Measurement of the diameter of the A549 and LC-HK2 homotypic spheroids on different days in the culture. Data are shown as mean relative to day 5 in three independent experiments (SD). Statistical analysis was performed using Dunnett’s test with an ANOVA test for multiple comparisons vs. day 5. * *p* value < 0.05 and ** *p* value < 0.001.

**Figure 3 cells-12-02790-f003:**
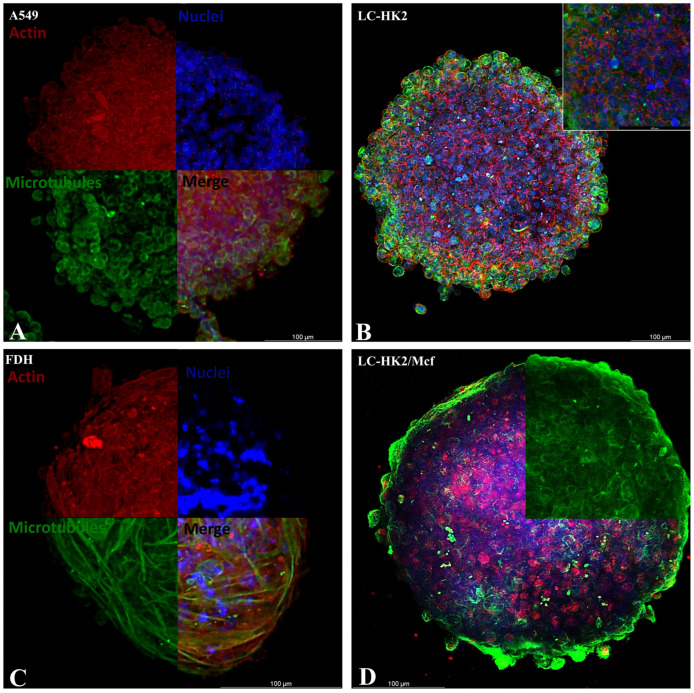
Confocal laser scanning microscope (LSM) images of actin filaments and tubulin microfilaments: image of the spheroids showing separate channels (actin and microtubules), and merging. Immunofluorescence of microtubules (green) of spheroid cells was performed with primary antibody anti-mouse and secondary antibody Alexa 488, the nucleus (blue) stained with DAPI, and the actin cytoskeleton (red). Cellular organization is evidenced by the actin cytoskeleton staining and mitotic cells, by microtubules in the mitotic spindle. Scale Bar: 100 µm.

**Figure 4 cells-12-02790-f004:**
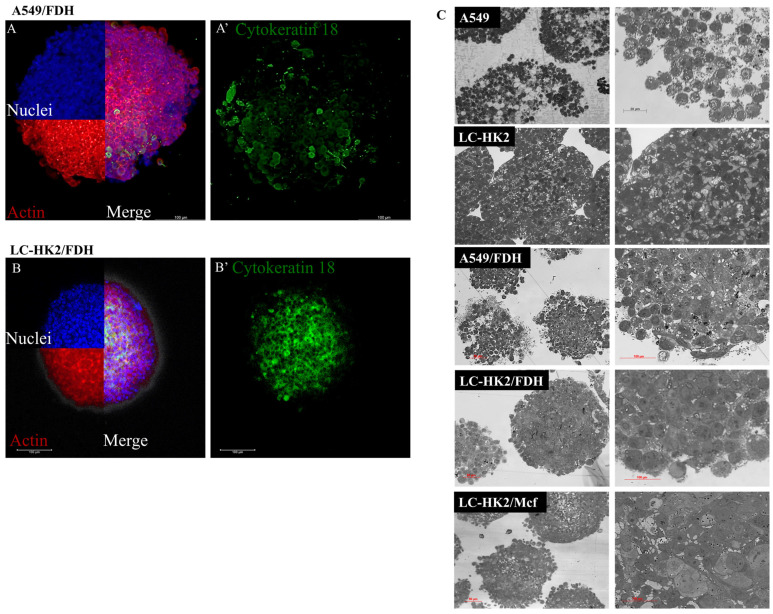
Morphological characterization of homotypic and heterotypic spheroids. (**A**,**B**) To visualize the cytokeratin 18 filaments within the heterotypic spheroids containing carcinoma cells, immunofluorescence was performed using an antibody anti-cytokeratin 18 mouse and anti-mouse alexa 488. Additionally, nuclei were stained with DAPI and with red actin cytoskeleton. Scale bar 100 µm. (**C**) Semi-thin sections (from TEM) of both homotypic and heterotypic spheroids. Scale bar: left, 50 µm and right, 100 µm.

**Figure 5 cells-12-02790-f005:**
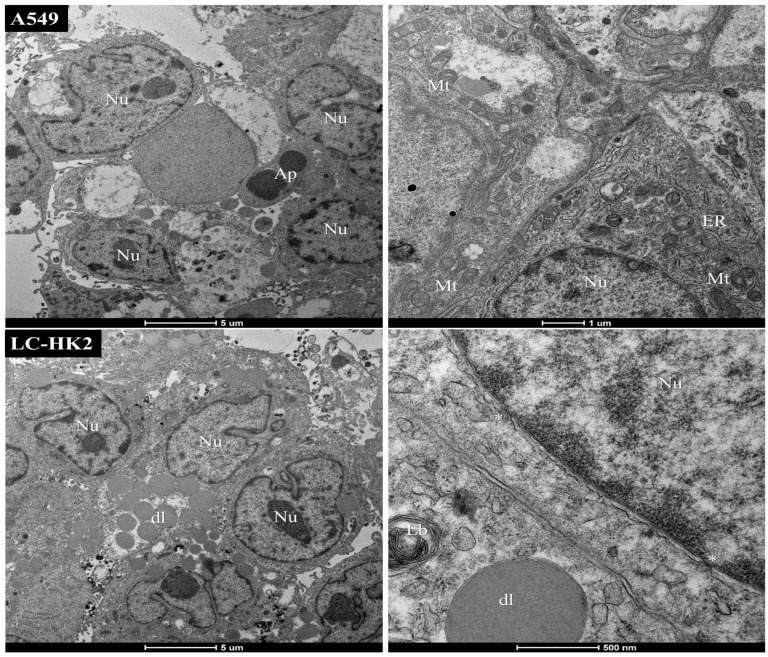
Transmission electron microscopy representative images of homotypic spheroids: cells from spheroids A549 and LC-HK2 have organelles such as nucleus (Nu); asterisks show nuclear pores (*); mitochondria (Mt), endoplasmic reticulum (ER), lipid droplets (dl), apoptosis (Ap), lamellar bodies (Lb).

**Figure 6 cells-12-02790-f006:**
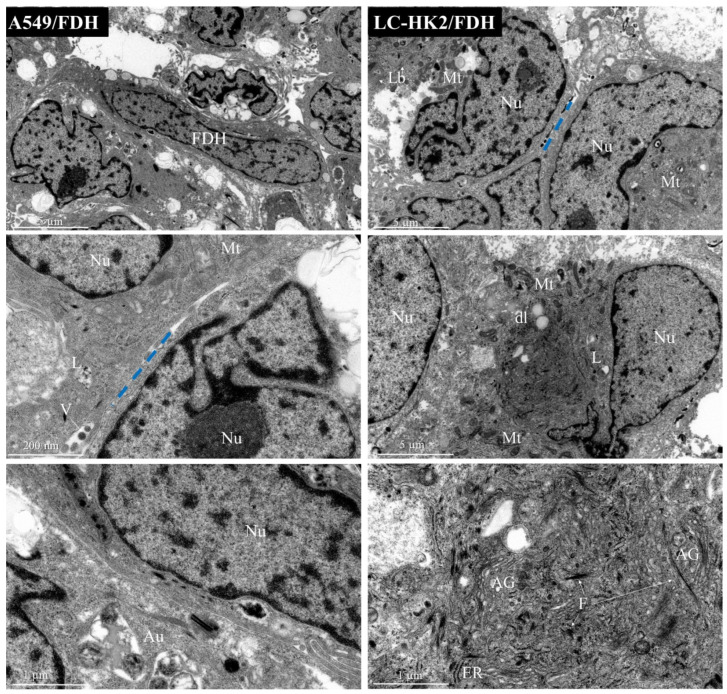
Transmission electron microscopy representative images of heterotypic spheroids with fibroblast: the cells of the spheroids A549/FDH and LC-HK2/FDH present organelles such as the nucleus (Nu), mitochondria (Mt), endoplasmic reticulum (ER), vesicles (V), lipid droplets (dl), lysosomes (L), autophagy (Au), Golgi apparatus (AG), lamellar bodies (Lb), filaments (F). Blue dotted lines show contact between two neighboring cells (----).

**Figure 7 cells-12-02790-f007:**
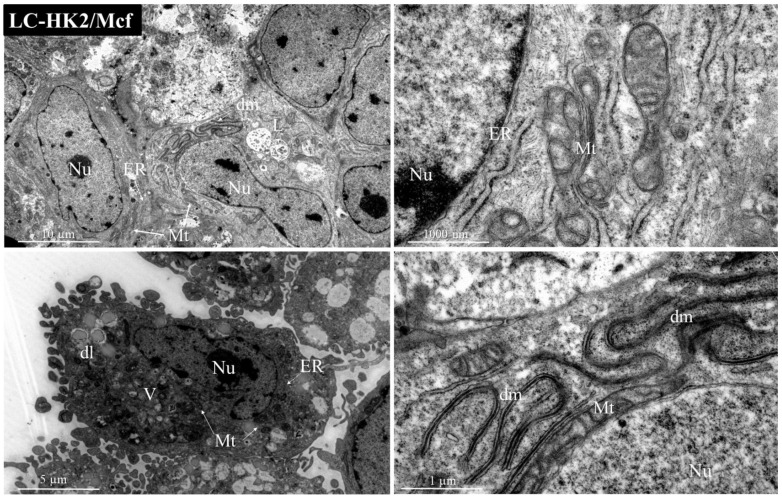
Transmission electron microscopy representative images of heterotypic spheroids with macrophages: LC-HK2/Mcf spheroid cells have organelles such as the nucleus (Nu), mitochondria (Mt), endoplasmic reticulum (ER), vesicles (V), double membranes (dm), lipid droplets (dl), lysosomes (L).

**Figure 8 cells-12-02790-f008:**
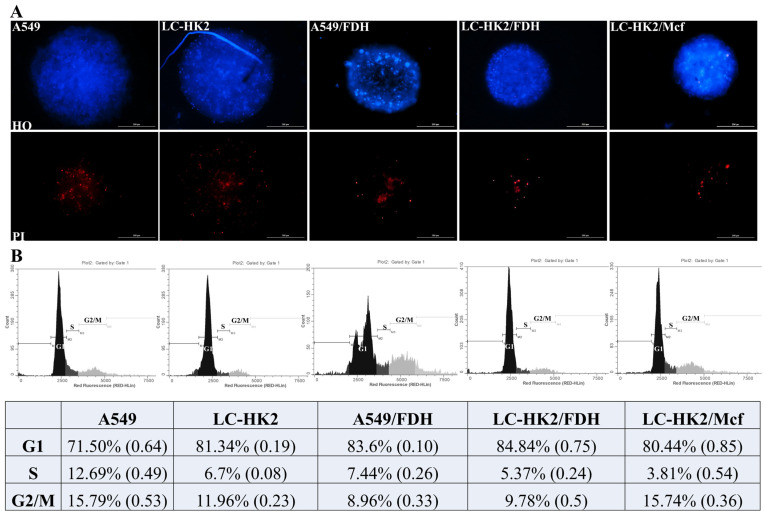
Viability and cell cycle of homotypic and heterotypic spheroids: (**A**) cell viability of spheroids with 7 days in cell culture, stained with Hoechst and propidium iodide; (**B**) cell cycle of spheroids with 7 days in cell culture. Data are shown as mean relative to control in three independent experiments (SD).

**Figure 9 cells-12-02790-f009:**
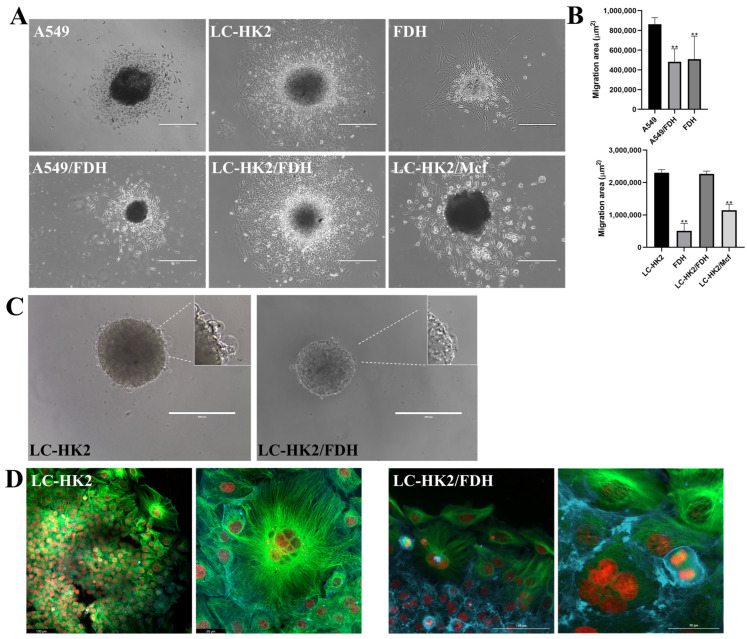
Characterization of spheroid cell migration. (**A**) Images after 72 h of spheroids placed on an adherent surface. Scale bar: 400 µm. (**B**) Measurement of the diameter occupied by the cells that migrated from the spheroids after 72 h; three spheroids per treatment were measured in triplicate. Statistical analysis was performed using Dunnett’s test with an ANOVA test for multiple comparisons vs. control. (**C**) Spheroids after 7 days of generation shown on their surface cells with much larger cytoplasm, in bubble format. Scale bar: 400 µm. (**D**) Immunofluorescence of cells that migrated from LC-HK2 and LC-HK2/FDH spheroids and nuclei were stained with IP (red), with microtubules in green and cytoplasm in blue. Scale bar LC-HK2 (image of laser scanning confocal microscopy): Right—50 µm, Left—100 µm; LC-HK2/FDH (image obtained by LionHeart microscopy): Right 30 µm Left—100 µm. ** *p* value < 0.001.

**Figure 10 cells-12-02790-f010:**
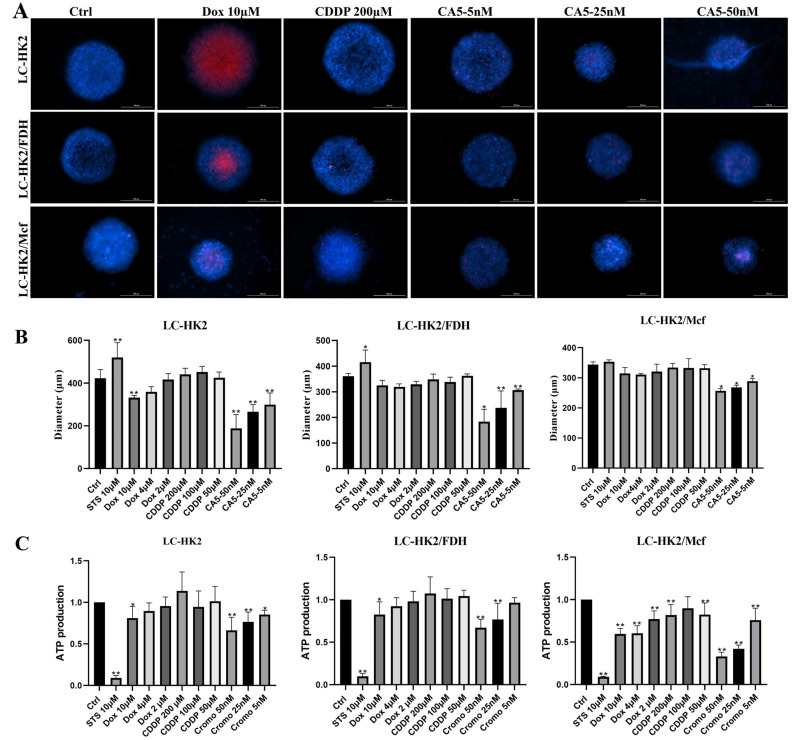
Differences in response to anti-cancer drugs of LC-HK2 heterotypic spheroids treated for 48 h. (**A**) Cell viability assay with HO and IP of spheroids treated for 48 h (merge). Blue cells represent spheroid-forming cells, and red areas represent non-viable cells. Bar Scale: 200 µm. (**B**) Measurement of spheroid diameter after 48 h of treatment. Three spheroids per treatment were measured in triplicate. (**C**) Measurement of ATP production of spheroids treated with Dox, CDDP, CA5 and Sts (used as a control for cell death). The measurements were normalized with respect to the control. The experiments were performed in triplicate. Statistical analysis was performed using Dunnett’s test with an ANOVA test for multiple comparisons vs control. * *p* value < 0.05 and ** *p* value < 0.001.

## Data Availability

The data presented in this study are available in article.

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
