# Peer review of "Characterization of 3D NSCLC Cell Cultures with Fibroblasts or Macrophages for Tumor Microenvironment Studies and Chemotherapy Screening"

_cells, 2023, doi:10.3390/cells12242790_

Round 1

Reviewer 1 Report

The authors studied cultivation and characterization homotypic and heterotypic 3D-models of non-small cell lung cancer. Such cellular models are important for study different interaction in tumor microenvironment for develop new antitumor treatment. After review of the manuscript by Garniqu and Machado-Santelli. I have the following comments as a peer reviewer assigned by the Office:

1.     The Introduction section would definitely benefit from being shortened and focused on aim of this study. For example, the history of cell culture and 3D cell models which is already well-known/well-studied, overloads the manuscript.

2.     In line 145 “1%, in DMEM/F12” looks misspelled.

3.     Is the FDH of fibroblast dermal culture? No abbreviations in the text.

4.     It is unclear at what cell ratio heterotypic cellular models were obtained and whether there is a dependence on the number of fibroblasts/monocytes added to form the models.

5.     It is not clear from the text of the manuscript whether spheroids formed from the three cell types for mimic the tumor microenvironment. The title of the article implies that a heterotypic model should consist of three cells components.

6.     In Fig. 2B no space between words.

7.     In Fig. 4 red signal is the actin cytoskeleton? For Fig. 4C white balance is not normalized.  

8.     Based on cytokeratin 18 alone, it is difficult to judge the location of cells in a heterotypic 3D model. On what day of cultivation was the experiment conducted? For example, using Cell Traсker, you can perform a quick test of the distribution of cell types in 3D during short-cultivation.

9.     In Fig.5-7 white text on gray background with the model’s name is difficult to read. In Fig.6 A549/FDH is overlaid on the bar.

10.  Whether the ratio of live to dead cells was assessed for spheroids of FDH. This type of cellular model is characterized by nemosis DOI: 10.1016/j.yexcr.2009.03.005 and Fig. 1 showed strong aggregation cells for FDH.

11.  In line 436 “staurosporine (10 mm)” looks misspelled.

12.  In Fig. 8 and 10 letter designations with a parenthesis.

13.  The Discussion section would definitely benefit from being shortened and more focused on findings of the study.

Regards, Reviewer

Author Response

We thank the reviewers’ suggestions very much. The manuscript modifications are in labelled in yellow.

And bellow we summarized the answers to reviewers’ observations;

Reviewer 1

  1. The Introduction section would definitely benefit from being shortened and focused on aim of this study. For example, the history of cell culture and 3D cell models which is already well-known/well-studied, overloads the manuscript.

-Introduction was shortened

  1. In line 145 “1%, in DMEM/F12” looks misspelled.

 This was in line 143. We couldn’t find the misspelling

  1. Is the FDH of fibroblast dermal culture? No abbreviations in the text.

It was only in the abstract, now it is included in Material and Methods

  1. It is unclear at what cell ratio heterotypic cellular models were obtained and whether there is a dependence on the number of fibroblasts/monocytes added to form the models.

Cell ratio was 1:1 with both cell types. Included in the text.

  1. It is not clear from the text of the manuscript whether spheroids formed from the three cell types for mimic the tumor microenvironment. The title of the article implies that a heterotypic model should consist of three cells components.

Only 2 cell types were used. Observation included in the text.

  1. In Fig. 2B no space between words.

Corrected

  1. In Fig. 4 red signal is the actin cytoskeleton? For Fig. 4C white balance is not normalized.  

Yes, it is in figure legend.

  1. Based on cytokeratin 18 alone, it is difficult to judge the location of cells in a heterotypic 3D model. On what day of cultivation was the experiment conducted? For example, using Cell Traсker, you can perform a quick test of the distribution of cell types in 3D during short-cultivation.

Please, see the paper:

Menz, A., Weitbrecht, T., Gorbokon, N. et al. Diagnostic and prognostic impact of cytokeratin 18 expression in human tumors: a tissue microarray study on 11,952 tumors. Mol Med 27, 16 (2021). https://doi.org/10.1186/s10020-021-00274-7

They concluded that “Our data show that CK18 is consistently expressed in various epithelial cancers, especially adenocarcinomas.”

  1. In Fig.5-7 white text on gray background with the model’s name is difficult to read. In Fig.6 A549/FDH is overlaid on the bar.

Corrected

  1. Whether the ratio of live to dead cells was assessed for spheroids of FDH. This type of cellular model is characterized by nemesis DOI: 10.1016/j.yexcr.2009.03.005 and Fig. 1 showed strong aggregation cells for FDH.

The induction of nemosis in fibroblast spheroids can´t be discarded. It seems not occur in heterotypic spheroids.

I

  1. In line 436 “staurosporine (10 mm)” looks misspelled.

“staurosporine (10 µm)”modified

  1. In Fig. 8 and 10 letter designations with a parenthesis.

Corrected

  1. The Discussion section would definitely benefit from being shortened and more focused on findings of the study.

Discussion was modified in order to attend both reviewers.

Please, find  the new version of the correted manuscript attached.

Reviewer 2 Report

This article investigates the use of 3D cell cultures to mimic the tumor microenvironment of non-small cell lung cancer (NSCLC) and evaluate the effects of chemotherapeutic and potential compounds on homotypic and heterotypic spheroids. The study generated both homotypic and heterotypic spheroids of NSCLC with fibroblasts or macrophages, which were characterized morphologically and analyzed for cell viability, cycle profiling, and migration. The results showed that different cell populations were identified based on spheroid zoning, and drug effects varied between spheroid types. The study concludes that the complexity of the spheroid should be considered when analyzing compound effects.

Strengths:

The research question is significant and novel.

The methodology is well-described and includes the generation of spheroids, analysis of metabolic activity using different assays, and evaluation of the response to different chemotherapeutic agents.

The data analysis is comprehensive and includes statistical tests to compare the results across different experimental conditions.

The article is well-organized and includes clear headings and subheadings to guide the reader through the different sections.

Weaknesses:

The article lacks a clear hypothesis or research objective that outlines the specific goals of the study.

The sample size for each experimental condition is not clearly stated, which makes it difficult to assess the statistical power of the study.

The discussion section is relatively brief and does not fully explore the implications of the findings or suggest avenues for further research.

Suggestions for improvement:

Clarify the research objective and hypothesis in the introduction section to provide a clear guide for the reader.

Clearly state the sample size for each experimental condition and justify the selection of the specific cell lines and chemotherapeutic agents.

Expand the discussion section to fully explore the implications of the findings and suggest potential avenues for further research.

Consider adding a limitation section to discuss potential shortcomings of the study and suggest ways to address them in future research.

Author Response

cells-2521463

We thank the reviewers’ suggestions very much. The manuscript modifications are in labelled in yellow.

And bellow we summarized the answers to reviewers’ observations;

REVIEWER 2
Clarify the research objective and hypothesis in the introduction section to provide a clear guide for the reader.

We  described the objectives in detail, please see in the text.

Clearly state the sample size for each experimental condition and justify the selection of the specific cell lines and chemotherapeutic agents.

Included

Expand the discussion section to fully explore the implications of the findings and suggest potential avenues for further research.

Discussion was modified in order to attend both reviewers’ suggestions.

Consider adding a limitation section to discuss potential shortcomings of the study and suggest ways to address them in future research.

 Included

Please, find  the new version of the correted manuscript attached.
